# Contemporary Collecting in a Pandemic: Challenges and Solutions for Documenting the COVID-19 Pandemic in Memory Organizations

**Chiara Zuanni**

Centre for Information Modelling, University of Graz, Elisabethstrasse 59/III, 8010 Graz, Austria; chiara.zuanni@uni-graz.at

**Abstract:** This paper draws on a survey of contemporary collecting projects aiming to document the COVID-19 pandemic in museums and other memory organizations. The survey focused on European memory organizations and investigated the practices they adopted in collecting, accessioning, recording, preserving, and displaying material and immaterial witnesses of the pandemic. This paper presents the results of the survey, with a particular attention on the challenges faced by these projects in relation to born-digital objects. It discusses how organizations were able to quickly launch initiatives aimed at collecting memories of the pandemic, drawing on different collection methods, adapting to the circumstances, and using a proactive attitude to reach out to different communities. The paper highlights the solutions adopted to ensure legal compliance in these projects and discusses the need for ethical considerations in relation to the collection of traumatic memories. It suggests that these collecting projects are likely to face significant challenges in the subsequent processing of this material—due to its volume and the need for new digital curation and preservation workflows. Therefore, the paper argues that these projects could also lead to a renewed attention and collaboration across the heritage sector on issues of digital curation and preservation.

**Keywords:** COVID-19; contemporary collecting; digital curation; rapid-response collecting

## 1. Introduction

This paper focuses on the practices enacted by memory institutions to collect memories of the COVID-19 pandemic, discussing how these projects are contributing to changes in contemporary collecting workflows and strategies. The scale at which material and immaterial witnesses of the pandemic have been collected by a range of institutions and communities around the world since 2020 is striking, with collecting projects being launched by museums, archives, libraries, web archives, and researchers in different fields (history, sociology, anthropology, contemporary archaeology, etc.). This paper argues that these projects, collecting everyday objects and narratives, have not only a direct interest for their documentary value in relation to contemporary history and heritage, but they also have emphasized the difficulties and challenges memory institutions encounter in collecting, curating, and interpreting the assemblages of physical and born-digital objects of a digital society.

Museums have always collected the contemporary, whether in social history collections, art and design, or science ones. In recent years, history, social history, and archaeology collections have increasingly acknowledged the digital character of late 20th and early 21st century society: from USB sticks found in excavations [1] to digital photography documenting historical events (e.g., 9/11), numerous digital objects are entering museum collections as witnesses of contemporary history and society. Contemporary digital photography sits in both historical and art collections, documenting events and experiences, and fostering participatory practices [2]. Digital art collections have an even longer experience of dealing with hybrid artworks, relying on a range of technologies (including

different hardware and software) that need to be maintained with attention to both the artists' original intent and the practices of digital preservation [3,4]. In parallel, design museums also have collected digital devices and software as documentation of contemporary design trends (e.g., videogames, iPods, and smartphones, etc.—see [5–7]). Since 2014, rapid-response collecting—a practice first proposed by the Victoria and Albert Museum—has also emerged as an approach to rapidly collect objects following events. Social media content has often been targeted by these projects either as a form of documentation and as a form of including different voices and perspectives in the collection [8,9].

However, by choosing to collect contemporary memories and objects, these projects are increasingly dealing with the challenges of a digital society, in which memories are often shared on digital platforms (e.g., social media), preserved in digital formats (e.g., smartphone photos), and constructed through online interactions (e.g., hashtags connected to live-events). The collection, acquisition, preservation, interpretation, and display of this born-digital material is challenging museum collecting practices and—despite an emerging body of research in this area—there is still a lack of guidelines and workflows in relation to these practices. The challenge of collection from contemporary witnesses, often in born-digital formats has been heightened during 2020, when many institutions set out to document the COVID-19 pandemic and its impact on society. These challenges have been further compounded by the situation in which many pandemic collections were initiated, during multiple lockdown periods, with the closure of public spaces, in a time of great uncertainty. The traumatic nature of the event being documented affected both collecting staff and donors, and legal and ethical challenges were heightened. Even when collecting physical objects, social distancing regulations required new approaches for the acquisition of such objects. Subsequently, the volume of these collections, and their hybrid nature (including physical and digital objects), represented another significant difficulty to their cataloguing and preservation, highlighting the need for appropriate digital preservation skills, infrastructure, and workflows in the museum sector.

The research presented here has investigated the different solutions museums adopted to face these challenges, and it stems from the project "Contemporary collecting and COVID-19: barriers, bottlenecks, and perspectives in digital curation", funded by a DARIAH Theme grant (2020–2021). The project aimed to foster discussion around the problems of contemporary collecting among professionals working in different memory institutions and to research practices, challenges, and possible solutions in relation to the data management and public dissemination of such collections. This paper draws on the results of a survey and short interviews, conducted in 2021, to explore collections of memories of the COVID-19 pandemic and the challenges museums and other organizations encountered in these projects.

In the following sections, after a short review of the challenges emerging in contemporary collecting and a presentation of the research methods, a discussion of the survey results on practices adopted by museums to (a) collect, (b) record, curate, and preserve, and (c) display memories and witnesses of the COVID-19 pandemic will follow. In doing so, this paper will contribute to the analysis of the impact of the pandemic on museums and to literature on contemporary born-digital collecting.

## 2. Contemporary Collecting and the Pandemic

Despite its longer history, which can be traced back to the beginning of the modern museum and has evolved particularly in the context of social history, science, and design collections, contemporary collecting (including its form as rapid-response collecting) has become an increasingly relevant and urgent theme within the museum community in recent years. It supports the broadening of museum narratives, the inclusion of under-represented communities, and co-curation initiatives, and it allows a quick response to evolving trends and events, including traumatic ones [10]. At the same time, however, contemporary collecting practices have to increasingly manage born-digital objects, such as photography or social media texts and images [2,11–13]. In these regards, museums are now facing

challenges well-known also in the web archiving and digital preservation community: the curation of born-digital data as contemporary heritage is therefore concerning to a range of memory institutions, raising both theoretical and methodological issues [14].

In short, the peculiar nature of contemporary collections, which are often assemblages of physical artefacts and born-digital material, poses many challenges in relation to: collection (methods of collection and accession in cultural heritage organization), curation (due to the sheer volume of objects often collected in such projects and the closeness to the events being documented), preservation (due to the lack of clarity on file formats, metadata standards, authoritative vocabularies, and sustainable storage solutions), display (due to the tensions between analogue and digital materials), ownership (rights management), ethics (due to the sensitive nature of some material and data protection implications), representation (in order to achieve a balanced representation of all society), and participation (encouraged through crowd-sourcing projects).

During the coronavirus pandemic, museums closed worldwide and, in many cases, increased their digital activities. UNESCO estimated that 90% of museums had been affected by temporary closures [15] (p. 13). Both ICOM and NEMO surveyed museum organizations, also evaluating the increased attention to digital methods during this period [16–19]. Despite differences in the development of digital initiatives, due to the capacity and possibilities of each organization, it is possible to identify some clear trends, such as an increase in online exhibitions (through a range of technologies), a move of public programs to online platforms (such as videoconferencing and streaming ones), a sustained and established use of social media, and an increase in contemporary collecting activities [20]. Indeed, the increase in contemporary collecting, with a focus on documenting the experiences of the pandemic, was not only notable within the sector but it also caught the attention of the media [21–23]. Kosciejew [24] revisited the history of contemporary collecting and emphasized the need for collecting memories of the pandemic, reflecting on the duty of memory organizations to document and preserve them.

Indeed, different types of organizations launched collecting initiatives: besides museums, archives, historical societies, research projects from anthropology, archaeology, public history, and digital humanities, as well as the social sciences, initiated collections of memories, in the form of tangible and intangible objects relating to the pandemic and its impact. These efforts have been documented in gray literature and reports, and there is an emerging body of academic literature. Most of these publications focus on single projects, describing well-delimited initiatives: for example, Chu [25] discusses the collection of Asian American stories at the Museum of Chinese in America (MOCA) in New York; and Laurenson, Robertson, and Goggins report on the work at National Museums Scotland [26]; Visintainer, Feldman, Kruger, and Livingston reported on the coordination between the different collecting projects at California State University [27]; Rodriguez documented how curators and archivists at the David J. Sencer CDC Museum collected the response of the Centers for Disease Control and Prevention (CDC), the organization leading the US response to the pandemic and therefore a crucial but also vast source of materials and information [28]; Nyitray, Reijerkerk, and Kretz described the methods, mainly oral histories, adopted to document experiences at Stony Brook University [29]; Schendel [30] described his experience and reflections on collecting in a time of crisis, based on the collection project at the Evansville Museum of Arts, History & Science. Emmens and McEnroe [31], drawing on the work they did at the Science Museum London, highlighted the urgency of collecting the often ephemeral objects related to the pandemic, such as face-masks and health-related messages and equipment, noting how these ephemera were characteristics also of the Spanish flu pandemic, but they were very scarce in museum and archive collections. Patterson and Friend [32] reflect on the implications of collecting children's memories of the pandemic, with attention to the ethical questions these collections pose, in particular in relation to the inclusion of children's voices in the documentation and interpretation of their experiences.

So far, a more limited body of literature has emerged considering the broad context of these projects collecting memories of the pandemic. Kamposiori [33] has conducted a short survey on the activities of Research Libraries UK (RLUK) organizations, noting how the participants started the collections early in the spring of 2020, with a focus on institutional and local memories and the aim of developing such collections for teaching and research purposes, as well as a way to engage with their communities. Tizian Zumthurm surveyed the scope of crowdsourced collections of the pandemic, drawing on a selection of projects to highlight content and method adopted, and identified the following subgroups: projects that "asked specific questions to a selected group of individuals", projects that collected pictures of specific events, projects focused on collecting the experiences of specific groups, projects launching a fully open call; furthermore, he adds a category driven by the platform, rather than the scope, of this project, i.e., projects using Omeka S [34]. Jones, Sweeney, Milligan, Bak, and McCutcheon survey the situation and approach in Canada, highlighting the need for considering the representativeness of the collection and of developing digital preservation capacity in memory organizations [35]. Spennemann [36] discusses the need for collecting broadly and intensely in the present and focuses then on the curation and preservation of these collections, proposing a framework for accessioning the objects in subsequent phases, so as to allow time to evaluate their relevance within the collections.

## 3. Materials and Methods

This paper draws on a research project that surveyed collections of memories and witnesses of the pandemic, with a focus on understanding the practices adopted in collecting and recording this material. In particular, this paper presents the results of a survey, which was conducted in the spring of 2021. The survey, available in English and German and targeted at European organizations, was promoted through social media, mailing lists, and by contacting directly relevant projects. It contained 35 questions investigating the context and focus of projects collecting memories of the pandemic (dates, geographical scope, thematic focus) and the processes that had been set up to collect the objects (physical and born-digital), record them (acquisition and cataloguing workflows, including considerations on formats and metadata for born-digital objects), conservation and preservation, and any eventual plan for the display of the objects. Legal and ethical issues raised both by the type of objects and the emerging impact of these experiences on the organization's collecting and digital strategies were also investigated.

Each section of the survey included a few multiple-choice questions, and a few open questions to allow the addition of details and a short description in each participant's own words. While the multiple-choice questions allow identifying broader trends, although with no statistical reliability given the self-selecting and limited number of answers, the open questions allow a more granular discussion of the different approaches and challenges encountered by memory organizations in documenting the pandemic. At the same time, these open questions also expand and justify the themes first investigated through multiple-choice questions, offering a clearer picture of the processes and practices by the different collecting projects.

The survey received 60 complete answers and some partial ones. In a few cases, follow-up interviews were organized with respondents who had agreed to be interviewed and left their contact details in the form (responding to an invite in the last page of the questionnaire). The questionnaire included a statement about the use of the data, and all the results are presented here as aggregated and anonymous. The data will be discussed mainly through a qualitative lens, following a summary of the situation as emerging from the quantitative analysis of the survey's responses. The discussion is complemented by an observation of the online presentation of European projects aiming at documenting at the pandemic and by reflections emerging from the interviews and events of this research project.

## 4. Results

The survey was circulated both in English and German, and this had a noticeable impact on the results, which include 18 respondents from Germany (9), Austria (8), and Switzerland (1). Overall, different European regions were well represented, including answers from Croatia, Denmark, Finland, France, Greece, Hungary, Ireland, Italy, Lithuania, Malta, Norway, Poland, Portugal, Sweden, and the UK.

Different types of institutions also appear well-represented, with participants selecting—as a descriptor for their organization—"Archaeology, anthropology, ethnology museums", "Archives", "Art gallery and museums", "Design museums", "History museums (including city and regional histories museums)", "Science Museums", "Social history museums", and a few other types of museums (contemporary and digital art, regional museums, folklore museums). In parallel, projects initiated by cultural associations and university researchers were also represented in the survey results. Most of the participants had experience in contemporary collecting projects preceding the pandemic, as part of social history collections, of contemporary art and design collections, or as part of born-digital collecting activities.

The majority of the projects documented in the survey started between March and April 2020, with the beginning of the first wave of the COVID-19 pandemic in Europe and its consequent lockdowns. Similarly, the majority of these projects (38) were still accepting new contributions in the spring/summer of 2021 and had not yet established a date for stopping their collection of memories of the pandemic. Only seven projects did not include any geographical boundary to their collection, declaring an international scope in their approach to the collection, whereas all the other projects had a national (17), city and/or local community (12), or regional (11) scope.

The focus of the collections presents some similarities, with many projects focusing on photographs, written memories, and representative objects (e.g., face-masks). In addition, newspaper cuttings, signs and oral testimonies from local businesses, drawings, questionnaires and diaries, posters, vials of the COVID-19 vaccine, and other health-care-related material. Different types of born-digital data were also included in these collections, such as Twitter archives, virtual live performances, municipality and/or government communication, recorded interviews, audio-visual testimonies, and oral histories. Though many of these initiatives were open to submissions from every member of the public, some were targeting from the beginning specific groups (e.g., frontline workers experiences, schoolwork, grocery-store workers, children, university students). Similarly, while there was a shared focus on everyday experiences during the pandemic and objects identified as symbolic and representative of this period, there were also some projects that chose to focus on specific stories and aspects of the pandemic, such as fear, objects that became symbols of companionship/feelings during lockdowns, and stories of solidarity.

The survey of the projects focused subsequently on the methods and workflows adopted by these projects to collect, manage, and preserve these collections of memories and witnesses of the pandemic, and the plans being considered for their future as well as the approaches to legal and ethical concerns emerging from this material. These processes are of particular interest because, as has been noted above, the collection of memories of the pandemic has notably contributed to the growth of attention to rapid-response collecting and the discussion on the methods and tools for the curation of contemporary collections, which tend to include hybrid assemblages of physical and born-digital objects. The following sections will therefore discuss how the survey participants dealt with the challenges of digital curation and how those experiences shaped their approach to collecting and digital strategies.

### 4.1. Collecting Memories of the Pandemic

A first set of questions in our survey investigated the methods adopted to develop collections of memories and witnesses of the pandemic, considering previous experience in contemporary collecting practices and the scope of the collections. A proactive and flexible

approach, drawing on a range of methods, in which online forms, social media, and—when possible—physical encounters to facilitate donations were all used during the different phases of the projects, accompanied by a sustained promotion and communication of these projects to foster participation.

The majority of the respondents did not have an established workflow for contemporary collecting projects, with a respondent describing their approach as based on 'spontaneous practices'. Others quickly setup a workflow, relying on their curatorial background to develop a set of practices targeted at their COVID-19 projects. Only three respondents reported that they had been developing workflows for contemporary collecting since before the pandemic: two of these referred specifically to the collection of born-digital images (thanks to their participation in the Collecting Social Photo project and their use of the Samtidsbild applications). The third respondent referred instead to the presence of a curator for contemporary history in their organization, as well as an expectation that every curator collected 'contemporary culture as a matter of routine', which had helped them to develop appropriate workflows over the years.

Independently from their previous experience, all of the projects relied on multiple collection methods: the most popular collection method was via an online form, either implemented on the organization's website or via a common survey platform (such as SurveyMonkey or Google Forms), but email submissions were also largely encouraged. Social media were widely used to promote the collecting projects and encourage participation, but only ten projects made social media hashtags and posts the primary focus of the collection. Furthermore, despite the participatory intent and crowdsourcing approach of many of these projects, existing crowdsourcing platforms were not mentioned—except for two projects, using Omeka, a content-management and publishing system, which is used also in crowdsourcing projects. Finally, some projects focused on targeting specific communities, such as medical and front-line staff or local shops and businesses, in order to collect their experiences and ephemera related to the pandemic. While all these methods led to the collection of born-digital objects, mainly in the form of texts or digital photographs, organizations also collected physical documentation of the pandemic—either by actively seeking those artefacts in their area or by encouraging members of the public to bring them to the institution (often after first contacting the organization via email). As could be expected, the time frame for the collection of the physical artefacts had been considerably influenced by the situation imposed by the pandemic, and there were still cases of objects that had not yet been collected physically due to the impossibility of safely contacting the donor.

Looking at the responses to the questions dealing with collection methods, it is noticeable how the variety and adaptability of collection methods on the one hand, and collaboration with key groups and the media was at the core of most projects. Curators had a proactive approach visiting shops and public spaces to document the visible traces of the pandemic and contacting stakeholders to ask for contributions (a respondent estimated that his/her organization had been writing on average 10 invitations per week to selected individuals and groups), as well as actively observing and crawling social media platforms. This variety of methods was actively sought so as to cover a range of experiences of the pandemic period, with a museum also encouraging the writing of short texts by visitors in its galleries. Furthermore, the collection methods also changed with the development of the COVID-19 pandemic, as in the case of a museum that set up a collaboration with a vaccination center in 2021. Collaboration and promotion in the media were considered a key factor in the development of these initiatives: thus, online web forms were paralleled by interviews, museum volunteers, and member associations acted as multiplicators for the initiatives expanding their reach, and the projects were promoted, both through social media and traditional media. A museum regularly changed the theme promoted on its social media for its collecting project, so as to highlight different topics every few weeks and thus encourage a range of diverse donations. The mention of these projects in the news, especially in local newspapers, radio, and digital news platforms, alerted

many potential donors and contributed to expanding participation. As one respondent wrote, there was "a conscious effort by us to share [the call for contributions], but it also became word-of-mouth".

In conclusion, despite different levels of previous experience in contemporary collecting projects, the projects had all to deal with an unprecedented situation, in which a combination of approaches, a proactive stance monitoring and being ready to cover gaps in the on-going collection, and on-going promotion of these initiatives were deemed crucial. The resulting collections included both physical and born-digital objects, the latter including texts, photos, and audio and video content, either submitted through ad hoc channels (online forms, emails) or collected on social media. Museums, archives, and research projects ended up therefore with rich collections, which in turn needed to be accessioned and catalogued. These collecting initiatives led to heterogeneous collections of physical and digital artefacts, which—given their content and character—challenged existing cataloguing practices. The next section will therefore observe how these projects contributed to highlighting challenges and testing solutions for the curation of contemporary memories and objects.

*4.2. Cataloguing and Preserving Memories of the Pandemic*

The management and curation of these collections of pandemic memories raised many challenges for the collecting organizations. Not every project planned to preserve all the material collected or had already made clear plans for the curation of the objects. While all the participants could rely on clear accession and cataloguing workflows and policies in their respective organizations, there were differences in the approach and choices in relation to these collections. Twenty-five respondents had already made clear plans for the accession and preservation of these objects, while twenty-nine had focused on the cataloguing of the collections. Conversely, some participants had not yet decided to which extent they were going to access and preserve the material or had not yet decided whether to catalogue it. Indeed, the approach of this group reflects the suggestion of Spennemann (2021) that a certain interval of time should be left between the collection and its evaluation and eventual accession, allowing more reflection on the long-term value of the collection itself.

As a respondent commented in the survey, the challenge was "digitising large collections of ephemera", i.e., recording and managing the large number of objects collected. More precisely, it seems that survey participants were dealing with three critical aspects in relation to the accession and recording of these objects: (1) time and staff capacity, (2) legal constraints, and the (3) the lack of practices for born-digital data.

First of all, the large volumes of memories and objects collected proved difficult to manage, especially in the context of the pandemic (home-offices, financial difficulties). In most cases, the project team was very small, if not an individual initiative, so that the entering of all the necessary information into the organization databases seemed to be a daunting task. This profusion [37] appeared as common to both physical and born-digital objects. In addition, some respondents mentioned that, due to the current situation, they still had to physically collect the objects from their donors (who had so far sent only digital images of the objects). Secondly, as it will be discussed in a further section, many respondents were dealing with sensitive material, which required careful ethical and legal consideration. In some cases, besides provisions taken from the beginning of the initiative, they also were evaluating how to access and record the material appropriately. Thirdly, respondents raised the challenge of dealing with new objects, born-digital ones, such as Instagram stories, Facebook posts, and digital images/videos. The preservation of this material, identifying the right formats in which to archive it; the comprehension of its context (within the broad realm of our digital lives) and date (e.g., in the case of memes or other popular posts), in order to ensure its conservation and understanding for the future were considered crucial. For a few institutions, this meant the beginning of born-digital collections, and they consequently were in the process of considering long-term strategies for such objects.

The survey also focused on the existence of preservation workflows for contemporary collecting and, in particular, born-digital collections within the organizations and the impact of these pandemic experiences on such practices. The vast majority of the participants referred to their preservation practices for physical objects (through their approaches to conservation and storage), and one respondent mentioned that their plan also included the printing and archiving of born-digital sources (e.g., emails printed and stored in a physical archive). Conversely, one respondent could draw on an existing workflow for the archival of born-digital material, while another organization reported on the existence of an internal set of guidelines for collecting digital assets.

It seems, therefore, that the respondents were by and large still relying on analogue preservation processes, although as a respondent commented, "more and more digital material changes the routine". It can be expected that these projects will represent a significant first experience towards the development of digital preservation workflows and, indeed, further questions led to the emergence of a challenging situation and a need for more refined practices for the curation of born-digital artefacts. First, we asked in what format the digital objects had been collected, and the answers ranged from media type, e.g., 'video', 'photographs', and 'documents' to more detailed lists of specific file formats, e.g., '.jpg', '.tiff', '.mp4', and '.pdf'. Secondly, a question asked explicitly if the respondents had planned or were implementing any kind of file conversion, so as to convert the donated files—often in different formats—to more optimal formats appropriate for long-term archival. In this case, only twelve respondents answered positively, whereas twenty-one answered negatively and fourteen chose the option 'I still haven't faced this challenge, but I might encounter it in the future' (an option designed to check awareness of the issues surrounding format, even if a plan had not yet been fully developed). A similar picture emerges from the observation of the metadata adopted for this material. In short, many of the projects seemed to lack sustainable digital preservation practices for born-digital material. Indeed, despite an increasing attention to the challenges of born-digital contemporary collecting in recent years, there are not yet comprehensive and shared solutions for memory organizations working with these objects, and it could be further argued that the experiences of collecting memories of the pandemic are going to raise further attention to the need of developing appropriate workflows and strategies for born-digital objects.

*4.3. The Future of the Collections*

Besides the impact these collections will have on the future of digital curation practices within the institution, it could be also questioned how these collections will be presented to the public. Questions on the plans for the display and presentation to the public of these objects revealed that thirteen organizations wished to display these collections in the future but had not yet decided the format, whereas seven were already planning a temporary exhibition in their spaces, nine planned to present the collections only online, and eleven did not have any plan for their display. The plans for dedicated exhibitions ranged from temporary exhibitions on aspects of everyday life during the pandemic (e.g., health measures, community, solidarity, humor) to exhibitions looking at the COVID-19 pandemic in comparison with previous pandemics (e.g., the Spanish flu).

Only in six cases had the collections already been presented to the public: three cases on a website and three cases in a temporary small exhibition. Interestingly, the three physical exhibitions had taken different forms: as a small addition at the end of a display on the history of the respective region, in a 'new acquisition' case, and in a focused temporary exhibition. It is also interesting that one respondent reported that they were not planning an exhibition in the immediate future, because they wanted to "give it at least minimal historical perspective [and] so [an exhibition] could be done in maybe 10–20 year after the pandemic is over". Indeed, other respondents also highlighted how their main interest was in documenting the pandemic for future generations rather than for preparing an

exhibition, and the question of a distance between the pandemic and its public presentation came up frequently also in interviews and discussions relative to the project.

### 4.4. Legal and Ethical Aspects

The collection of memories of the pandemic included legal and ethical challenges, given the nature of the collections and the contexts of these initiatives. On the one hand, all of the organizations had consistent measures in place to guarantee respect for copyright and GDPR legislation, as well as national and institutional policies. As mentioned above, the most popular collecting method relied on online forms, and legal departments (when present) had advised on the requirements and phrasing of the forms. Common solutions adopted during the collecting process included anonymization, licensing (in a few cases using Creative Commons licenses), and the redaction of any other personal data, with follow-up questions posed to the donors when necessary. In the case of social-media-based collections, only one respondent—coming from the web archiving community—acknowledged the compliance to platforms' terms and conditions, whereas other respondents who used social media to collect memories of the pandemic considered the data public and did not mention any process with regard to the terms of the platforms and their use as archives. It seems therefore that legal aspects were carefully considered, in line with a museum's usual acquisition processes, with the notable exception of social media content, which was not fully recognized as in need of specific legal considerations.

Conversely, the attention to ethical aspects—in relation to the potentially sensitive and traumatic nature of the material for both staff and donors—seemed to have received less attention. Fewer organizations had a clear strategy from the beginning, with most of the respondents testifying to an evolving approach. On the one hand, most staff had voluntarily chosen to be involved in these projects, and there was a broad acknowledgment of the need for regular breaks during work with pandemic-related objects and memories. It seems that the nature of the projects, drawing on rapid-response initiatives, did not allow time to plan in more detail the long-term impact on the staff, in terms of workload and potential triggering material. As Schendel notes, this experience led him to shift his approach to rapid-response collecting and to highlight the mental health impact on museum staff during these projects, which—he argues—should become a factor in deciding whether to develop or not such collecting initiatives (2021). I would also suggest that, given the uncertain development of the pandemic, this lack of support might have contributed to the gradual decrease of collecting activities in 2021, with many projects limiting their collection to the spring and summer of 2020. On the other hand, the participation of the public was also on a volunteer basis, and it seems that donations were mostly focusing on the first changes in everyday life at the beginning of the pandemic, highlighting the novelty of the lockdowns and other measures rather than the most tragic aspects. In addition, many organizations had included links and contact references for mental health organizations and resources in their online forms and planned to include such contacts also in any eventual exhibition.

To conclude this section, it should be noted how legal and ethical considerations affect all the processes, from the collection to acquisition, preservation, and display of this material—in particular the born-digital one. The projects represented in the survey had carefully considered their compliance with current legislation and regulations, although the collection of social media data and the impact on the staff of such projects seemed to have been less consistently planned. This appears to be due to the nature of these projects, drawing on rapid-response practices in which the urgency of collecting prevailed. However, many projects acknowledged the need for reflection on the consequences on staff and audiences of dealing with the trauma associated with these collections and their development during a challenging period. Furthermore, as noted in the previous sections, the recording of these collections and plans for their display were still at an initial stage, and it could be expected that during these processes, the organizations will have to refine and strengthen legal and ethical considerations for this type of contemporary, often born-digital, collections.

## 5. Discussion

This paper has presented the results of a survey of contemporary collecting projects aiming to document the COVID-19 pandemic. In doing so, it has emphasized common themes, challenges, and solutions memory organizations faced in collecting, curating, and displaying material and immaterial witnesses and memories of the pandemic.

All of the projects surveyed here had quickly and efficiently launched different initiatives aimed at collecting both physical and born-digital objects, relying on a variety of complementary collection methods. However, due to the ongoing pandemic, the lack of clear practices for managing born-digital objects, and the sheer volumes of these collections, most projects were also facing difficulties in processing the material they had collected. On the one hand, they faced curatorial choices in the accession of the objects, in order to ensure a manageable and representative collection. On the other hand, the digitization and recording of physical artefacts and long-term archival solutions and the recording of born-digital objects required significant time and effort, as well as novel solutions. As could perhaps be expected, even fewer plans had already been developed for the display of the artefacts. Legal issues had been considered throughout the projects, often with the support of the relevant legal departments in the organizations. Conversely, ethical issues, due to the sensitive and traumatic nature of the event document, i.e., the COVID-19 pandemic, but also to the situation in which such collections were developed, had been faced with less preparation and with an approach evolving and adapting in relation to the circumstances of staff and donors.

The experiences of the collections of memories and witnesses of the pandemic discussed in this paper, with many organizations facing the difficulties of born-digital objects for the first time, have contributed to further highlight current challenges in contemporary collecting projects—and, at the same time, the values of these projects. Indeed, contemporary collecting, also in its form as rapid-response collecting, has grown in interest in recent years. Many organizations turned to it during the pandemic so as to both document this period and to foster a continuing relationship with their audiences through a participatory approach to the collection. In doing so, they had to adapt to circumstances in order to safely collect physical and born-digital objects; to proactively monitor the on-going collections, so to address any potential gap in its representation and in the inclusion of different experiences; and to deal with the potential traumatic and triggering nature of many of the memories and objects collected. The new collecting workflows, such as the online forms for contribution set up in these projects; the ability to quickly pivot to a range of participatory collection methods and promote them to different communities; and a better awareness of the ethical safeguards needed to support staff carrying out these projects will likely facilitate and support new rapid-response initiatives.

At the same time, research in recent years has been focusing on the contemporary challenges collecting faces in a digital society. As discussed, the documentation of current events and trends needs indeed to also account for their online personal and collective memorialization, which generates a variety of born-digital objects. Yet, there is a lack of workflows and standards that support the collection, curation, and mediation of born-digital content in memory organizations. As evidenced from the survey results, it is likely that many projects will struggle to record and preserve the material collected, which often consists of different file formats and requires different metadata for its description. In addition, the storage and digital preservation of this content is a long-term challenge for these collections. There is therefore a need for more exchanges and reciprocal support between the web archiving and digital humanities community on the one side and museum and heritage practitioners on the other side, in order to develop the capacity for working with contemporary collections, consisting of physical and born-digital objects.

To conclude, I would argue that the collection of memories of the pandemic has not only constituted a significant documentation of the COVID-19 pandemic for the future, but it has also the potential for resulting in renewed attention and networking around issues of digital preservation and curation across the heritage sector, memory organizations,

and digital preservation specialists. This research has highlighted some of the practices successfully adopted by collecting projects in this period, as well as some critical questions. In particular, I have suggested that these experiences have led to the development of successful collecting practices that could relatively easily be drawn upon in future projects. However, the following steps, including the acquisition, cataloguing, and preservation of these collections, still present considerable challenges. Thus, more research and collaboration will now be needed to further develop resources to support rapid-response initiatives and—crucially—improve infrastructures and standards for the long-term preservation of hybrid and born-digital collections in memory organizations.

**Funding:** The research and survey were funded as part of the DARIAH Theme 2020 Call.

**Institutional Review Board Statement:** Ethical review and approval were waived for this study due to the type of questionnaire and its target (addressing professionals with questions specific to a project, without collecting any data on the respondents themselves).

**Informed Consent Statement:** Informed consent was obtained by all survey participants, with a consent statement presented at the beginning of the survey.

**Data Availability Statement:** Data available on request due to restrictions. The data presented in this study are available on request from the corresponding author. The data are not publicly available due to privacy issues (the non-anonymity of the participants).

**Acknowledgments:** The author acknowledges the financial support by the University of Graz and by DARIAH EU. I am also grateful to all the participants in the survey and in the workshops.

**Conflicts of Interest:** The authors declare no conflict of interest.

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
