# Peer review of "Contemporary Collecting in a Pandemic: Challenges and Solutions for Documenting the COVID-19 Pandemic in Memory Organizations"

_heritage, doi:10.3390/heritage5040188_

Round 1

Reviewer 1 Report

This is a well-researched paper sharing important results of a survey on COVID-19 documentation by cultural heritage institutions. Strangely, it seems to neglect libraries in general, but mentions museums and archives as well as other disciplines.

19 - "contributing to changes in contemporary"

34- acknowledged (misspelled)

36 - "numerous digital objects entering museum"

46/47 - "following events" cut "quickly emerging trends".

58 - "the challenge of collection from contemporary witness often in born-digital formats has been heightened during 2020, when many institutions set out to document"

61 - "compounded by multiple public space lockdown in a time"

63 - "required new approaches for the acquisition"

78/80 - "research methods, discussion of the survey ... COVID-19 pandemic will follow"

89 -  "quick response"

102 - "preservation"challenges mentioned  include metadata, controlled vocabulary and archival formats; but this does not make sense either in the field of physical or digital preservation. Challenges should include file formats, storage, sustainability in digital preservation.

111- "temporary closures. Both ICOM..."

121 - "need for collecting memories"

123 - End sentence at "memories". Delete following starting at "as well as the ephemerality...epidemic"

126 - delete also

127 - "the social sciences"

128 - "memories, such as tangible and digital objects"

133 - "report"

187 - " questions to allow addition of details and description in each participant's own words."

201 - "following a summary"

211 - "and the UK"

213 - "institutions also appear well-represented"

218 - "parallel, projects initiated by cultural associations and university researchers were also represented."

235 - " newspaper cuttings", signs, and oral testimonies, drawings"

249 - "manage, and preserve these"

285-  "ten projects made a primary focus of the collection of social"

288 - Omeka is not necessarily a crowd-sourcing platform?

292 - "physical documentation of the pandemic"

338 - "organizations. Not every project"

361- " This profusion appeared as a common challenge to"

364 - "as it will be discussed"

382 -  "printing and archiving"

384 - "stored in an preservation" not archival collection

397 - " - to more optimal formats"

421 - "in six cases had the collections already been"

461 - cut the sentence starting "I would also..." Resume with The participation of the public"...

474 - " the born-digital phase. The projects represented in the survey"  

479 - "the need for reflection on the consequences"

522 - " Cut "I" statement and start with ... "The collection of memories..."

524 - "resulting in renewed attention" not renovated

Author Response

First of all, I am very grateful for your careful and helpful suggestions to improve the language and style of my paper. I have implemented all of them (except in a couple of points, where I rewrote the full sentence, e.g. in the case of Omeka). 
In response to your comment on a lack of discussion on libraries: I agree they were important institutions that launched many interesting projects and I mentioned them in the literature section (in particular, Kamposiori's report on RLUK libraries). However, since the survey was targeted at museums, the results were limited in regards of libraries, and I did not want to risk generalisation, having too little data and I would prefer to discuss them in a separate future work, eventually. Thanks again for the many helpful corrections!

Reviewer 2 Report

Thank you for the opportunity to review this manuscript, which is very well written and makes important points about rapid-response collecting around the COVID pandemic. Since much of the manuscript deals with the collection or creation of documentary heritage, the author could have gone further in the literature on digital archives. Additionally, the framing of the issue in terms of rapid-response collecting makes a lot of sense. I would have liked to have seen the author draw out the implications of their study for rapid-response collecting more generally, and not just for rapid-response COVID collections, in the Discussion. Since we live in an era of crises, including a global pandemic, the ongoing climate crisis, with democracy being eroded and threatened around the globe, surely we can anticipate an ongoing need for national and international rapid-response collecting. I would appreciate it if the author could include reasonable conclusions they can draw about how institutions can better prepare for this kind of collecting, so that we are not always reliant on "spontaneous practices." (6)

Author Response

First of all, I am grateful for your positive comments and very helpful suggestions.
I have expanded the discussion section, so to include more reflections on the implications of the contemporary collecting experiences of the pandemic for the future - in particular, highlighting the need of more collaboration with digital archives and preservation research. However, given the word limits and focus of this article, I have not added a longer discussion of digital archives and literature in this area - and I would instead propose such discussion in a future work. Thanks again for your helpful suggestions!

Reviewer 3 Report

This is a wonderful study and extremely timely, given that we are only seeing the implications of the pandemic on memory organizations such as museums and archives.  I feel that the only area where improvements could help is by making the relationship between your thesis and what the survey data shows more explicit in your abstract and introduction.  The abstract seems to focus on the survey but it should incorporate the essence of your thesis, which is what the data supports.  The conclusion is somewhat abrupt; instead, it may benefit from reflecting on your thesis, survey data, analysis, and discussion, which are excellent.  So, my review focuses on the structure, not methodology and the command of English -- no concerns in those areas.    

Author Response

First of all, many thanks for your nice words and your suggestions! I have now expanded the abstract and added a section in the introduction, making more explicit my thesis. I have also largely rewritten and expanded the conclusion, so to include a broader discussion. 
Thanks a lot for pointing out this need for clarity in the structure!